# Robotic-Assisted Medial Unicompartmental Knee Arthroplasty Provides Better FJS-12 Score and Lower Mid-Term Complication Rates Compared to Conventional Implantation: A Systematic Review and Meta-Analysis

**DOI:** 10.3390/jpm14121137

**Published:** 2024-12-03

**Authors:** George M. Avram, Horia Tomescu, Cicio Dennis, Vlad Rusu, Natalie Mengis, Elias Ammann, Giacomo Pacchiarotti, Michael T. Hirschmann, Vlad Predescu, Octav Russu

**Affiliations:** 1Department of Orthopaedic Surgery and Traumatology, Kantonsspital Baselland, 4101 Bruderholz, Switzerland; natalie.mengis@ksbl.ch (N.M.); elias.ammann@ksbl.ch (E.A.); michael.hirschmann@unibas.ch (M.T.H.); 2Faculty of General Medicine, Carol Davila University of Medicine and Pharmacy, Bvd. Eroii Sanitari Nr. 8, Sector 5, 050474 Bucuresti, Romania; horia.r.tomescu@stud.umfcd.ro; 3Faculty of General Medicine, “Iuliu Hațieganu” University of Medicine and Pharmacy, Str. Victor Babeş Nr. 8, 400012 Cluj-Napoca, Romania; cicio.dennis@elearn.umfcluj.ro; 4Faculty of General Medicine, University of Medicine, Pharmacy, Sciences and Technology “George Emil Palade” of Târgu Mureş, 540139 Târgu Mureş, Romania; rusu.vlad-nicolae.20@stud.umfst.ro (V.R.); octav.russu@umfst.ro (O.R.); 5Department of Anatomy, Histology, Legal Medicine, and Orthopaedics, Sapienza University of Rome, Piazzale Aldo Moro, 5, 00185 Roma, Italy; giacomo.pacchiarotti@uniroma1.it; 6Orthopaedics and Traumatology Department, Ponderas Academic Hospital, 014142 Bucharest, Romania; vlad.predescu@reginamaria.ro; 7Orthopaedic and Traumatology Department, University of Medicine, Pharmacy, Science, and Technology, 540139 Targu Mures, Romania

**Keywords:** robotic-assisted unicompartmental arthroplasty, conventional unicompartmental arthroplasty, patient-reported outcome measures, follow-up, meta-analysis

## Abstract

**Background**: Robotic-assisted unicompartmental arthroplasty (rUKA) is gradually gaining more popularity than its conventional counterpart (cUKA). Current studies are highly heterogenic in terms of methodology and the reported results; therefore, establishing the optimal recommendation for patients becomes less straightforward. For this reason, this meta-analysis aims to provide an up-to-date evidence-based analysis on current evidence regarding clinical outcomes and complication rates following rUKA and cUKA. **Methods**: A meta-analysis was conducted following PRISMA guidelines. Five databases were searched, PubMed via MEDLINE, Epistemonikos, Cochrane Library, Web of Science, and Scopus. The relevant inclusion criteria were as follows: comparative clinical studies in which medial rUKA was compared to medial cUKA (prospective or retrospective designs), (2) human studies, (3) meta-analyses for cross-referencing, and (4) English language. The relevant extracted data were patient demographics, patient-reported outcome measures (PROMs), range of motion, and complications. A random-effects meta-analysis and subgroup analysis were conducted. The results include mean differences (MDs) and odds ratios (ORs), along with 95% confidence intervals (CIs) for continuous and binary variables, respectively. **Results**: rUKA showed a higher overall FJS-12 score compared to cUKA, with MD = 6.02 (95%CI: −0.07 to 12.1), *p* = 0.05. At 6 months postoperatively, the MD increased to 10.31 (95%CI: 5.14 to 15.49), *p* < 0.01. At a minimum 36-month follow-up, cUKA had a higher all-cause revision rate, with OR = 3.31 (95%CI: 1.25 to 8.8), *p* = 0.02, and at a minimum 60-month follow-up, a higher aseptic loosening rate, with OR = 3.86 (95%CI: 1.51 to 9.91), *p* < 0.01, compared to rUKA. **Conclusions**: rUKA provides better FJS-12 results compared to cUKA, as well as lower all-cause revision and aseptic loosening rates at 36- and 60-month follow-up, respectively. However, long-term follow-up is still pending.

## 1. Introduction

Robotic-assisted unicompartmental arthroplasty (rUKA) is gaining popularity as robotic surgery expands in orthopedics [1,2]. The high theoretical implantation accuracy of rUKA could make it a valuable addition to the operating room [1,3], especially since implantation accuracy in UKA is closely linked to postoperative outcomes and revision rates [4]. However, the adoption of new technologies is often limited by costs and the need for technical expertise, requiring significant time and resources from healthcare providers and manufacturers. Additionally, institutional support is often lacking due to the severe cost pressure that global health systems are currently facing. The impact of surgeon volume is also becoming more prominent, as it can affect surgical time, postoperative complications, and implant costs, which companies often consider in their pricing [5,6]. Together, these factors limit the adoption of new technologies, particularly for younger and low-volume surgeons.

Evidence-based approaches are increasingly important in answering clinically relevant questions such as the outcome differences between robotic and conventional surgery [3,7,8,9]. This includes comparisons of patient-reported outcome measures (PROMs) [6,10,11,12,13,14], postoperative pain levels [7,15,16,17,18], and complication rates related to different implants or techniques [1,4,5,9,19,20,21,22,23]. Specifically, the potentially added clinical benefit of rUKA compared to cUKA is of increased interest, particularly as cUKA often yields excellent mid- to long-term results [24]. However, the literature is still inconsistent, making it difficult to determine whether rUKA offers significant advantages over cUKA regarding long-term PROMs, pain levels, and complication rates [3,10,11,13,25,26,27].

The primary reason for this uncertainty is the overwhelming predominance of observational, retrospective studies with short follow-up periods, in addition to few prospective studies and even fewer randomized clinical trials [13,16,17,28,29,30,31]. Consequently, drawing definitive conclusions about the impact of robotic-assisted surgery in UKA is challenging, leaving the value of the substantial investment in robotic technology uncertain.

Another crucial factor contributing to the substantial heterogeneity in the literature is the variety of patient-reported outcome measures (PROMs) used to assess specific, objective (Ex. Knee Society Score) and subjective joint (Ex. FJS-12) and patient state (Ex. Oxford Knee Score, EQ-5D-L) following surgery [4,9,15,17,18,26,32]. Unfortunately, none of the existing instruments provide a comprehensive evaluation of function, quality of life, and subjective or objective complaints [32] while at the same time adequately accounting for associated comorbidities. As a result, current studies report varying findings—some indicate significant differences, while others show no difference—in terms of patient-reported outcomes following rUKA compared to cUKA [1,4,25,33]. These discrepancies may be influenced by factors such as patient selection, surgical techniques [34], psychological variables [19], and pain management or rehabilitation strategies [33].

In the light of these considerations, the current study employs a systematic review and meta-analysis methodology to provide a comprehensive overview of postoperative patient-reported outcome measures—PROMs, pain levels, and complications—following rUKA compared to cUKA.

## 2. Materials and Methods

### 2.1. Information Sources and Eligibility Criteria

A systematic review and meta-analysis protocol was registered with PROSPERO, CRD42024595082, and conducted according to the PRISMA guidelines. Five databases were searched on the 24 March 2024 (see Appendix A for the search strategy). The database search included (1) MEDLINE via PubMed, (2) Epistemonikos, (3) Cochrane Library, (4) Web of Science, and (5) Scopus.

The inclusion criteria were as follows: (1) studies comparing rUKA and cUKA with reported clinical outcomes and complication rates (prospective or retrospective designs), (2) human studies, (3) meta-analyses for cross-referencing, and (4) English language. Studies reporting on navigation results, cadaveric studies, editorials, commentaries, surgical techniques, letters to the editor, or study protocols were excluded. The entire search strategy can be found on PROSPERO, CRD42024595082.

### 2.2. Selection Process

Initially, 2774 articles were identified, and after duplicate removal, a total of 1606 papers were included in the selection process. Four reviewers conducted the selection process, first by performing title, abstract, and finally full-text screening. The 1606 records included for title screening were split in half and two reviewers were assigned to each half. Inter-rater agreement was calculated and is reported as ICC with a 95%CI, as shown in Figure 1 for title and abstract screening. Disagreements were solved by discussion between the four reviewers. During the cross-referencing step, 21 meta-analyses and systematic reviews were identified. From these, we collected all comparative studies focusing on rUKA and cUKA and compared them with the studies previously identified in our literature search. No additional studies were found that met the inclusion criteria. As a result, a total of 19 studies were included for final data extraction.

### 2.3. Data Collection Process and Data Items

The 19 included studies were split in half and two reviewers were assigned for data collection for each half. The following general data were collected: (1) authors, (2) level of evidence, (3) demographics—patient age, sex, BMI, follow-up, and Kellgren–Lawrence and ASA grades, (4) surgery time, and (5) length of hospital stay. Clinically relevant data points were as follows: (1) pre- and postoperative patient-reported outcome measures (PROMs) including pain, (2) range of motion, and (3) revisions and reasons for revision.

### 2.4. Statistical Analysis and Synthesis Methods

The mean and standard deviation (SD), median, and count data were collected, when appropriate. The mean difference (MD) together with 95%CIs was reported for continuous variables, while odds ratios (ORs) together with 95%Cis were reported for binary variables. A random-effects model meta-analysis was used for all forest plots due to the expected significant between-study heterogeneity and variability, the small sample size of the studies, and the presence of outliers. Given these reasons, a random-effects model ensured generalizability of the overall effect while at the same time maintaining flexible statistical model assumptions. The heterogeneity of the overall effect was assessed by computing the I^2^ estimate. The following thresholds for heterogeneity were used [35]: (1) 0–40%, heterogeneity is negligible; (2) 30–60%, moderate heterogeneity; (3) 50–90%, substantial heterogeneity; and (4) 75–100%, considerable heterogeneity. When *p*-values were reported for demographics, the *p*-value was calculated using a two-tailed significance test of the unstandardized mean difference, accounting for unequal group variances.

Subgroup analyses were conducted when patient-reported outcome measures (PROMs) and complications were reported for specific follow-up periods. Mean differences were calculated for continuous variables, while odds ratios were used for binary variables. A random-effects meta-analysis was applied in each instance. Each subgroup forest plot indicates the minimum follow-up period in its title and the type of robot used under “Robot Type” label, whereas forest plots that combine all follow-up periods provide information on the specific follow-up period being reported instead of the type of robot used.

When the sample mean and/or standard deviation were not reported, the transformation described by Wan et al. was used to derive mean and/or standard deviation values from the median, range, and/or interquartile range that were available [36]. Regarding revision reporting, not all common revision reasons [37] were specifically reported in the included studies; many times, only the total number of revisions was identified. For these reasons, revision analysis was performed on four main categories: (1) total number of revisions, (2) revision to total knee arthroplasty (TKA), (3) revision for persistent pain, and (4) revision for aseptic loosening. Furthermore, a large number of revision cases and their specific etiologies were reported in studies that also included lateral UKA and therefore were excluded from the analysis [38].

## 3. Results

A total of 2446 patients were included. In total, 1172 patients received rUKA while 1274 patients received cUKA. Relevant patient demographics like age, gender, hospital stay, BMI, follow-up period, Kellgren–Lawrence OA grade, and ASA grade are reported in Table 1.

Preoperative patient status as described by KSS, overall health status (EQ-5D-L), pain levels, and range of motion showed statistical differences between the two groups and are reported in Table 2.

Postoperative ROM is depicted in Figure 2, revealing that early range of motion showed no significant differences between the two groups, with an overall MD = −0.18° (95% CI: −11.17 to 10.8), *p* = 0.97. As the follow-up progressed, at 6 months postoperatively, a significant MD in terms of range of motion could be identified, with MD = 2.5° (95% CI: −0.05 to 5.05), *p* = 0.05. At 24 months postoperatively, the mean difference in range of motion between the two groups normalized, showing no statistical difference, with MD = 0.47° (95% CI: −3.25 to 4.18), *p* = 0.8. Lastly, when all follow-up timepoints were taken together, the overall mean difference in range of motion between the two groups was MD = 1.05° (95% CI: −0.84 to 2.95), with an unsignificant *p*-value of *p* = 0.28.

Postoperative results in terms of patient- and clinical-reported KSSs are presented for multiple timepoints in Figure 3. At 6 months postoperatively, there was no statistically significant mean difference in terms of the KSS between the two groups, with MD = 5.79 (95% CI: −9.55 to 21.14), *p* = 0.46. The same effect was observed at one year postoperatively, with MD = −1.32 (95% CI: −14.51 to 11.87), *p* = 0.84. At 24 months postoperatively, the mean difference between the two groups was again not significant, with MD = −2.70 (95% CI: −11.12 to 5.73), *p* = 0.53. The same trend could be seen at 60 months postoperatively, with MD = 0.82 (95% CI: −17.99 to 19.64), *p* = 0.93. When all timepoints were taken together, there was no statistically significant mean difference between the two groups, with MD = 0.25 (95% CI: −5.16 to 5.66), *p* = 0.93.

Postoperative mean differences in terms of overall health status, as depicted by the EQ-5D-L score, are presented in Figure 4. This was not significant at 6 months after either robotic-assisted or conventional UKA implantation, with MD = 0.49 (95% CI: −0.05 to 1.04), *p* = 0.07. At 12 months, the MD seemed to decline even further to 0.28 (95% CI: −0.27 to 0.83), *p* = 0.31. Moreover, 24 months after implantation, the MD between robotic-assisted and conventional implantation was 0.36 (95% CI: −0.18 to 0.91), *p* = 0.19. At 60 months postoperatively, the mean difference between the two groups in terms of overall health status was still unsignificant, with MD = −0.48 (95% CI: −1.21 to 0.25), *p* = 0.20. Overall, when all follow-up timepoints were taken together, the mean difference between the two implantation methods was still unsignificant, at 0.25 (95% CI: −0.04 to 0.54), *p* = 0.10.

Postoperative pain following either rUKA or cUKA is reported at different follow-up timepoints in Figure 5. At 6 months following surgery, there was no significant mean difference between the two, with MD = −0.97 (95% CI: −2.93 to 1), *p* = 0.33. At 24 months following surgery, the two implantation methods were still comparable in terms of pain, with a statistically unsignificant MD = 0.31 (95% CI: −0.11 to 0.73), *p* = 0.15. Overall, when all studies were combined, no statistically significant differences between the two implantation techniques could be identified in terms of postoperative pain, with an overall MD = −0.53 (95% CI: −1.51 to −0.46), *p* = 0.29.

Postoperative results in terms of FJS-12 scores are presented in Figure 6. Compared to the previously presented results, patients who underwent rUKA implantation had a statistically significant improved FJS-12 score at the 6-months follow-up, with MD = 10.31 (95% CI: 5.14 to 15.49), *p* < 0.01. When all follow-up points were considered, the FJS-12 score of patients who underwent rUKA was still higher than for patients who received cUKA, with MD = 6.02 (95% CI: −0.07 to 12.11), *p* = 0.05.

When revision rates of implanted UKAs were compared, there was no statistically significant odds ratio identified between rUKA and cUKA, with OR = 1.68 (95% CI: 0.84 to 3.35), *p* = 0.14. When the subgroup analysis was performed for follow-up time periods, higher odds of revisions were detected for cUKA at 24 months postoperatively compared to rUKA, with OR = 1.93 (95% CI: 0.99 to 3.76), *p* = 0.05. Furthermore, for studies reporting revision rates at a minimum of 36 months, again statistically significant odds of revisions could be identified in the cUKA group, with OR = 3.31 (95% CI: 1.25 to 8.80), *p* = 0.02. Figure 7 shows the results for total revisions for all reported follow-up timepoints, as well as the total revisions up to 12 and 36 months.

Figure 8 depicts the odds for revisions to TKA from both rUKA and cUKA. When all follow-up timepoints were taken together, significant odds of revising cUKA to TKA were observed compared to rUKA, with OR = 2.48 (95% CI: 1.21 to 5.09), *p* = 0.01. A number of research papers have reported revisions to TKA at multiple timepoints in the first 12 and 36 months of follow-up. Both time frames identified statistically significant odds of revising cUKA to TKA at 12 months, with OR = 2.76 (95% CI: 1.32 to 5.76), *p* = 0.01, and at 36 months, with OR = 3.61 (95% CI: 1.49 to 8.76), *p* < 0.001.

Frequently, revisions of UKA following long periods of persistent pain are reported. Figure 9 summarizes all papers that report such revisions. Specifically, the overall revision rate of UKA due to persistent pain was not statistically different between the two groups, with OR = 1.02 (95% CI: 0.36 to 2.87), *p* = 0.97. At the 12-month follow-up, the revision rate was not statistically significant, with OR = 1.36 (95% CI: 0.42 to 4.30), *p* = 0.62.

Finally, Figure 10 summarizes the odds of encountering aseptic loosening in the two implantation groups. Overall revisions due to aseptic loosening in the cUKA group did not prove to have higher odds than in the rUKA group, with OR = 1.30 (95% CI: 0.47 to 3.56), *p* = 0.61. Nonetheless, at the 60-month follow-up, here defined as mid-term, revision odds due to aseptic loosening were detected to be significantly higher in the cUKA group compared to the rUKA, with OR = 3.86 (95% CI: 1.51 to 9.91), *p* < 0.001.

## 4. Discussion

The most relevant finding of the present study is that rUKA provides higher FJS-12 values compared to cUKA at all follow-up timepoints. However, only a few studies report this specific outcome measure, and therefore, the overall effect shows considerable heterogeneity. Additionally, the FJS-12 and KSS assess different aspects of patient-reported symptoms. The FJS-12 concentrates solely on joint awareness, while the KSS addresses multiple factors related to patient satisfaction, pain, and function. The KSS includes indirect terms like “bother” (e.g., “How much does your knee bother you during each of these activities?”), which can be interpreted as more subjective compared to the straightforward focus of the FJS-12 on joint awareness. As a result, the findings highlight a contrast between the significant differences observed in the FJS-12 and the insignificant differences in the KSS, EQ-5D-L, and pain across all follow-up timepoints between rUKA and cUKA. This suggests that patient satisfaction and overall joint-related symptoms, as measured by the KSS, can be divided into two main domains, joint function and pain, which are directly linked to the osteoarthritic process, and joint awareness, which specifically relates to the reconstruction of the joint plane [1,2,20,21,23,24,56,57,58,59] and proprioception. Specifically, while the differences in postoperative patient-reported outcome measures (PROMs) between the two procedures remain controversial [4,24,27,60], the accuracy of implantation and the restoration of joint-related biomechanical parameters appear to significantly favor robotic implantation [61]. Nonetheless, it is important to note that while PROMs provide a useful overview of a patient’s overall satisfaction with the procedure, the variables they capture are often influenced by comorbidities, expectations, and lifestyle factors [9,32,62]. These influences may limit their ability to provide a comprehensive assessment of postoperative joint function, particularly when surgeons focus exclusively on objective data and joint-specific parameters [57] or surgical performance [6].

rUKA has a better ROM at the 6-month follow-up compared to cUKA, but this effect becomes negligible at the 24-month follow-up and also dissipates when all follow-up periods are combined. While this result may be appealing to patients looking for a quick return to sports or activities, the mechanism behind this effect is unclear, especially since it diminishes after six months of follow-up [18,26,28,49]. Additionally, the impact of postoperative pain management and rehabilitation is poorly understood, as most studies do not address these factors or specifically investigate their influence on range of motion. Interestingly, postoperative pain management appears to play a significant role in achieving an early increase in range of motion. This is particularly evident when haptic boundaries are used, as postoperative inflammatory markers tend to be much lower compared to those seen with cUKA [7].

Furthermore, rUKA provides significantly lower all-cause revisions in the 36-month follow-up with negligible heterogeneity that could already be identified in the current literature, suggesting that the studies are likely measuring the same underlying effect and that the variability between studies is not enough to affect the overall conclusion.

Moreover, revisions from UKA to TKA are more prevalent in cUKA than rUKA, but the overall heterogeneity, although reported to be negligible in the present study, can be misinterpreted as supporting the validity of the overall effect. Given the overall low number of studies reporting on this specific revision pattern, future studies must better define reasons for revision and failure modes that lead to revision [37].

Aseptic loosening at mid-term follow-up (minimum 60-month follow-up) is more frequent in cUKA than in rUKA; however, there is negligible heterogeneity, which is due to the fact that only a few studies report the 60-month follow-up, with an overwhelming weight of a single study [39]. For this reason, future studies should specifically focus on clarifying the modes of failure of UKAs and providing well-designed cohorts that, ideally, would reduce the risk of bias to a minimum and accurately capture the outcome of interest. Specifically, it seems that if medial UKA fails due to aseptic loosening, it tends to fail in the first 60 months. This aspect is relevant because it provides an interesting perspective on bearing surface wear and its link to aseptic loosening, especially in the case of fixed-bearing cohorts [29,63,64].

This meta-analysis offers a comprehensive evaluation of the overall effect and utility of rUKA implantation. An important limitation is that the vast majority of studies employed a retrospective design, with significant overall risk of bias. That said, the existing literature is still lacking both in the number and the rigor of studies. Consequently, many of the presented forest plots have revealed substantial or considerable heterogeneity among the studies, indicating that the underlying effects may not be consistently examined across all articles, raising questions on the validity and reliability of the overall effect. At the same time, this aspect also raises questions about the validity of combining results, particularly when reporting on PROMs and revisions to TKA. Additionally, it is essential to understand and clearly discuss the phenomenon of negligible heterogeneity, as it can be misleading. For example, a negligible heterogeneity value of 0% in this study may occur when there are too few studies to assess the outcomes of interest, when a single study disproportionately influences the overall effect estimate, or when small sample sizes lead to significant overlap in confidence intervals, creating a false impression of low variability.

rUKA potentially provides better FJS-12 values and lower revision rates compared to cUKA. Nonetheless, moderate-to-substantial heterogeneity was present for the majority of the investigated overall effects. For this reason, future studies should be focused on understanding the impact of a patient starting and adhering to the rUKA procedure. Starting and adhering to rUKA are distinct from simply being assigned to the procedure, as most patients must also follow specific pain management and rehabilitation protocols after surgery. Therefore, future studies should account for biases arising from confounding factors and participant selection at the outset. At the intervention level, studies need to address potential biases related to the misclassification of interventions. Post-intervention, authors should consider biases due to deviations from the intended interventions, missing data, outcome measurement errors, and the selection of reported results. Lastly, comparative studies with follow-up periods longer than 60 months and more transparent reporting of revision reasons are needed to evaluate the long-term impact of rUKA on complications.

## Figures and Tables

**Figure 1 jpm-14-01137-f001:**
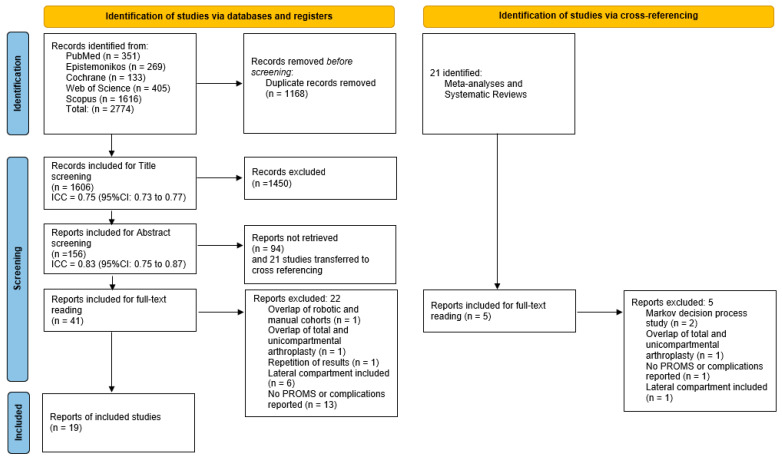
PRISMA flow diagram.

**Figure 2 jpm-14-01137-f002:**
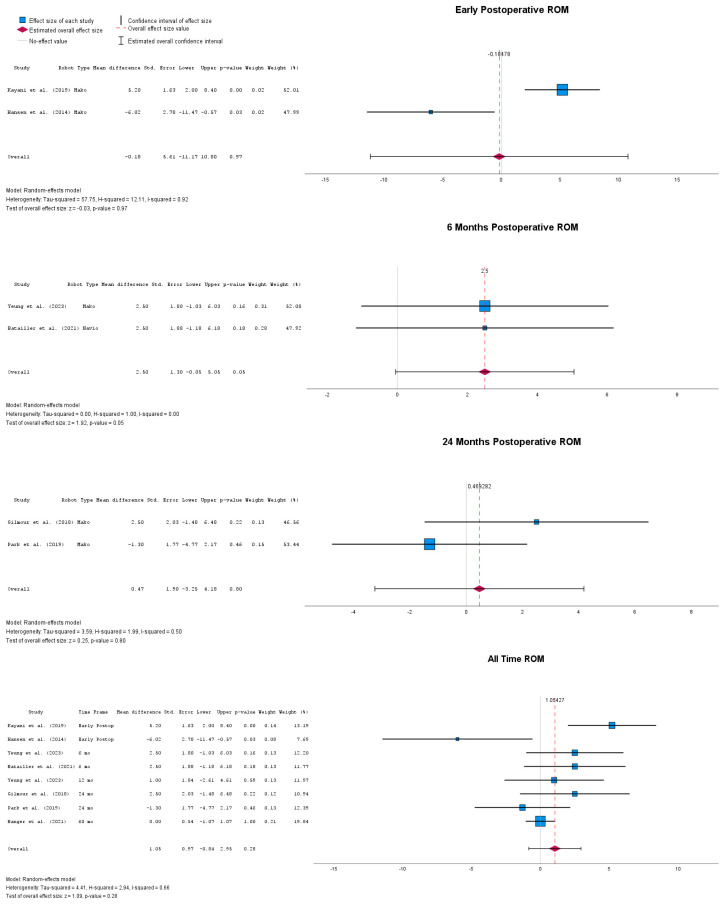
Forest plots of range of motion at multiple postoperative timepoints [33,44,45,47,48,50,51].

**Figure 3 jpm-14-01137-f003:**
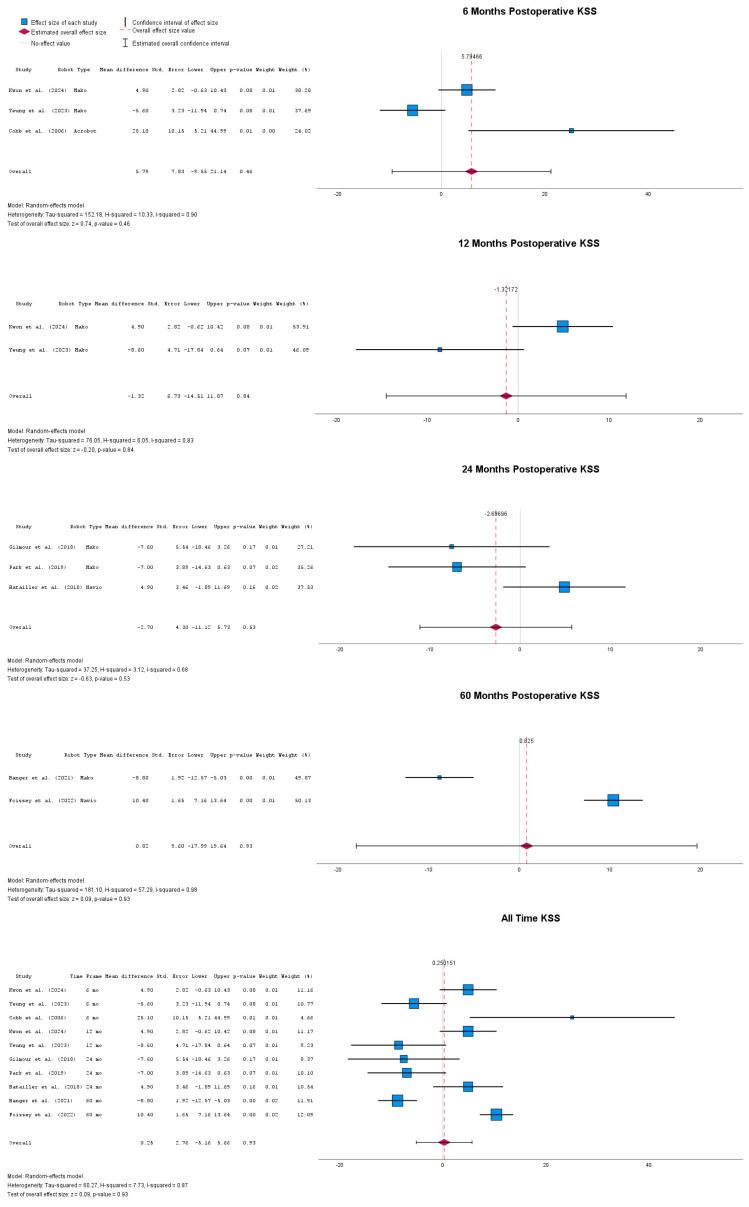
Forest plots of postoperative KSS at multiple postoperative timepoints [24,39,42,43,45,47,48,51].

**Figure 4 jpm-14-01137-f004:**
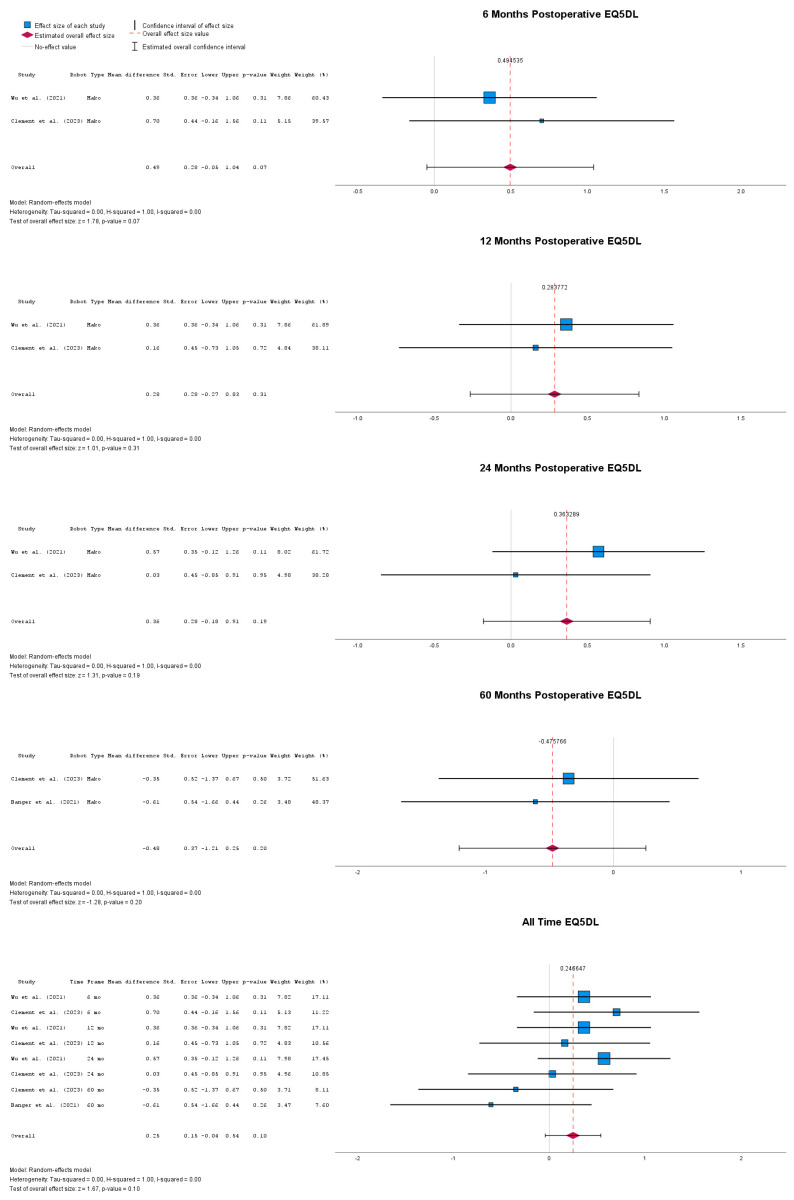
Forest plots of postoperative overall health status improvement depicted by EQ-5D-L score [40,41,48].

**Figure 5 jpm-14-01137-f005:**
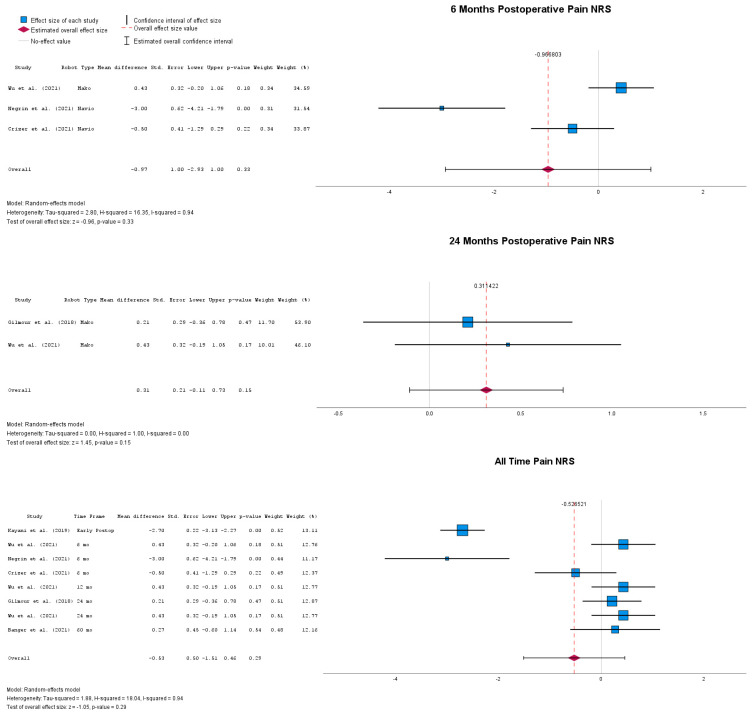
Forest plots of pain NRS at multiple postoperative timepoints [33,40,49,51,54].

**Figure 6 jpm-14-01137-f006:**
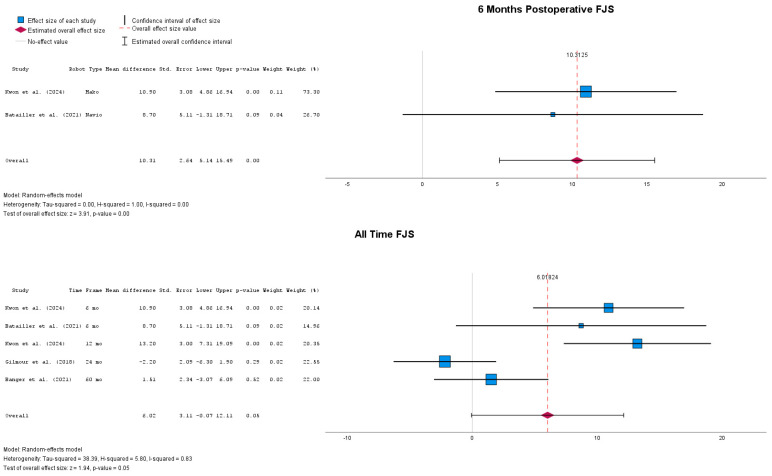
Forest plots of postoperative FJS-12 values [24,44,48,51].

**Figure 7 jpm-14-01137-f007:**
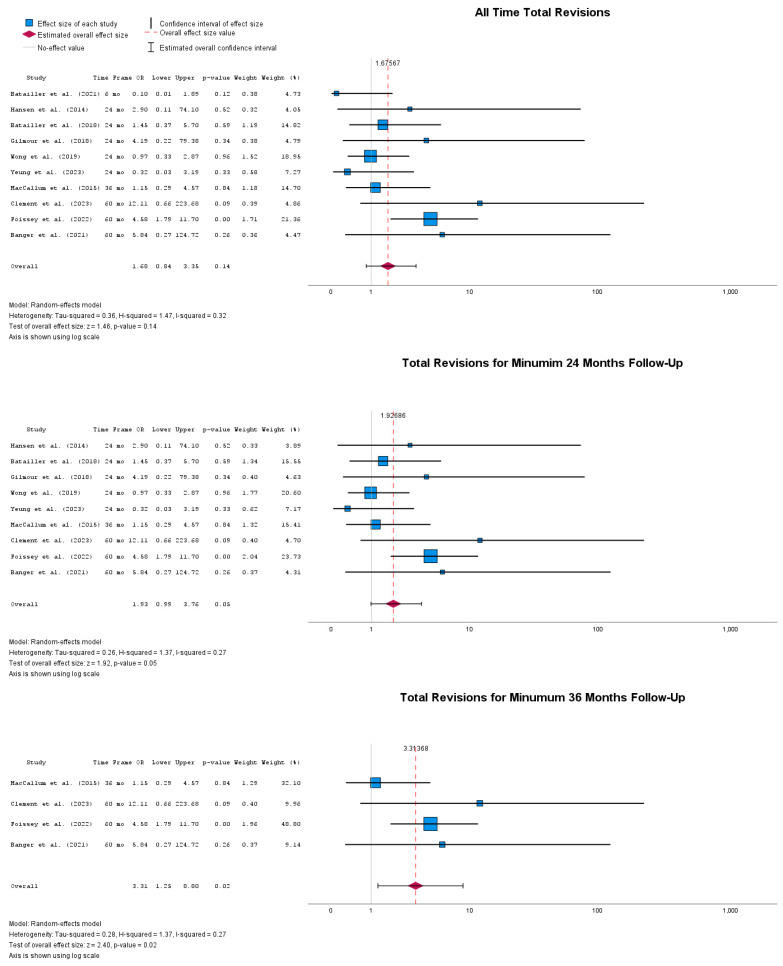
Forest plots of all time revisions at multiple timepoints [39,41,43,44,46,47,48,50,51,52].

**Figure 8 jpm-14-01137-f008:**
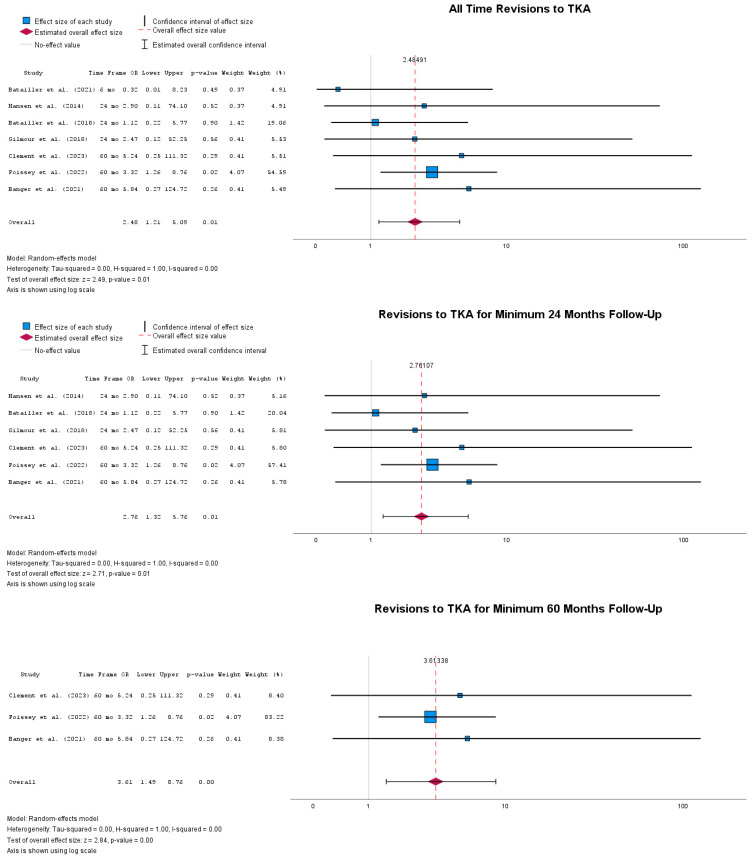
Forest plots of revisions to TKA at different timepoints [39,41,43,44,48,50,51].

**Figure 9 jpm-14-01137-f009:**
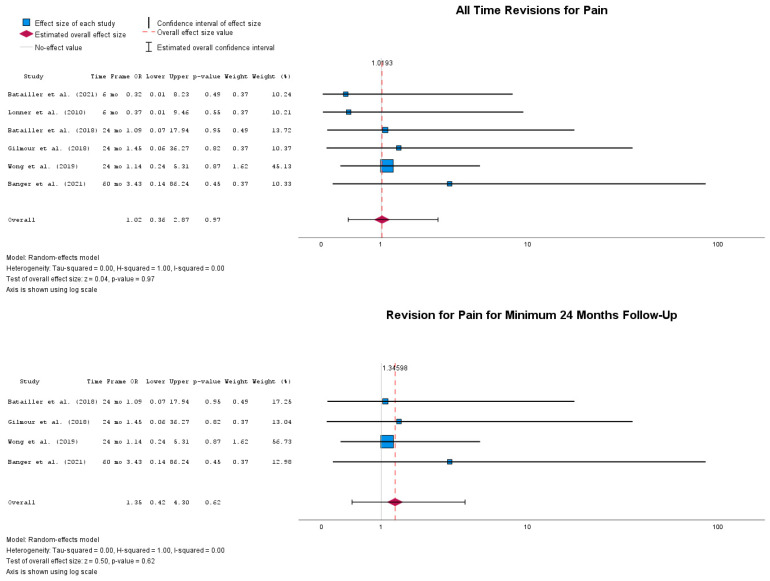
Forest plot of revisions due to persistent pain at different timepoints [43,44,46,48,51,52].

**Figure 10 jpm-14-01137-f010:**
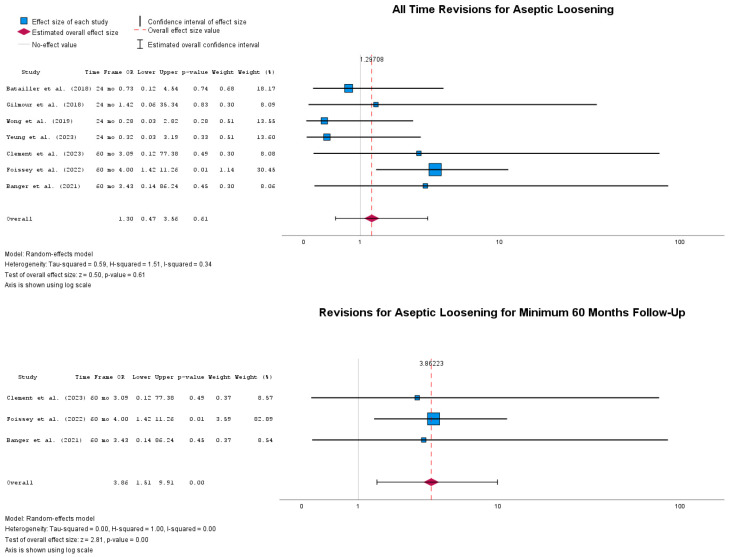
Forest plots of revisions due to aseptic loosening at different timepoints [39,41,43,47,48,51,52].

**Table 1 jpm-14-01137-t001:** Demographics of both manual and robotic-assisted groups.

	Manual	Robotic	Mean Difference	*p*
Total [24,33,39,40,41,42,43,44,45,46,47,48,49,50,51,52,53,54,55]	1274	1172	-	-
Age [24,33,39,40,41,42,43,44,45,46,49,50,51,52,53,54]	65.7 (8.9)	64.6 (8.9)	−0.26	0.77
Gender M/F (%) [24,33,39,40,41,42,44,45,46,49,50,51,52,53,54]	40.89/59.11	47.39/52.61	-	-
Hospital Stay [33,54]	2.8 (0.6)	1.9 (0.4)	−0.49	0.5
Operative Time [39,40,42,47,50,53,54]	73.7 (20.4)	85.7 (16)	11.94	0.04
BMI [24,33,39,43,44,45,46,49,50,52,53]	27.8 (3.9)	28.0 (4.1)	0.39	0.05
Follow-Up Period (Months) [24,33,39,40,41,42,43,44,45,47,50,51,52,53,54,55]	31.5 (19.57)	26.3 (9.46)	−8.44	0.09
Kellgren–Lawrence (0/I/II/III/IV) (%) [24]	0/0/0/62.86/37.14	0/0/0/54.29/45.71	-	-
ASA (I/II/II/IV) (%) [33,40]	24.63/66.42/0.75/0	28/68/0.8/0	-	-

BMI = body mass index; ASA = American Society of Anaesthesiologists physical status.

**Table 2 jpm-14-01137-t002:** Preoperative reported PROMs, pain, and range of motion in both rUKA and cUKA groups.

	Manual	Robotic	Mean Difference	*p*
KSS [24,39,42,45,47,51]	116.1 (18.5)	124.8 (18.7)	3.06	0.4
EQ-5D-L [40,41]	8.0 (9.9)	6.5 (2.7)	−0.89	0.59
Pain NRS [24,40,49,51,54]	6.3 (1.6)	5.7 (2.1)	−0.44	0.24
ROM [24,45,47,51]	120.6 (13.6)	119.0 (12.2)	−0.99	0.42

KSS = Knee Society Score; NRS = numeric rating scale; ROM = range of motion.

## Data Availability

All data are available on reasonable request.

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
