# Peer review of "Robotic-Assisted Medial Unicompartmental Knee Arthroplasty Provides Better FJS-12 Score and Lower Mid-Term Complication Rates Compared to Conventional Implantation: A Systematic Review and Meta-Analysis"

_jpm, 2024, doi:10.3390/jpm14121137_

Round 1
Reviewer 1 Report
Comments and Suggestions for Authors
This systematic review and meta-analysis compares robotic-assisted (rUKA) and conventional unicompartmental knee arthroplasty (cUKA) concerning clinical outcomes and complication rates. The study's main findings highlight that rUKA leads to higher FJS-12 scores, suggesting better knee function and reduced mid-term complication rates, specifically in aseptic loosening and revision surgeries, compared to cUKA. However, the results reveal no significant differences between rUKA and cUKA in overall health status (EQ-5D-L) and pain scores, indicating a nuanced benefit profile for robotic-assisted procedures. The manuscript underscores the potential advantages of rUKA but also reveals substantial heterogeneity in current literature due to diverse study designs and patient populations, limiting the generalizability of findings. Major revisions are proposed below to enhance the clarity, rigor, and relevance of the manuscript.
Comments and Suggested Revisions
-
Abstract - Objective: The objective could be clearer. Consider revising to "This systematic review and meta-analysis evaluates clinical outcomes and complication rates of robotic-assisted (rUKA) versus conventional unicompartmental knee arthroplasty (cUKA)."
-
Abstract - Results: Clarify “higher FJS-12” by adding context, such as “higher FJS-12 scores, indicating improved knee function,” to provide clearer information for non-specialists.
-
Abstract - Mid-term Follow-up: Define "mid-term" explicitly, as this term may vary. Specify follow-up duration both here and in the Results and Discussion sections.
-
Introduction - Clinical Relevance: Emphasize the unique clinical advantages of rUKA over cUKA, particularly in precision and joint alignment, to substantiate the study's significance.
-
Introduction - Cost Considerations: Mention how cost limitations impact patient access to robotic surgeries and may affect the adoption of rUKA.
-
Introduction - Evidence Gaps: Acknowledge the lack of long-term data comparing rUKA and cUKA outcomes to support the rationale for the study.
-
Introduction - PROMs Explanation: Define PROMs more comprehensively as "patient-reported outcome measures," clarifying what aspects of patient health they capture.
-
Methods - Study Selection: Expand briefly on the inclusion criteria, such as “Studies comparing rUKA and cUKA with reported clinical outcomes and complication rates,” to ensure clarity on study selection.
-
Methods - Database Search: Provide full registration details for the study protocol in PROSPERO.
-
Methods - Statistical Analysis: The choice of a random-effects model warrants clarification. Suggest briefly explaining why this model was used, particularly due to heterogeneity among study designs and populations.
-
Results - Demographics: The demographics in Table 1 lack sufficient context. Include a summary of key demographic findings within the text to facilitate reader understanding.
-
Results - PROMs and Complications: Suggest separating the discussion on PROMs and complication rates for better readability, rather than combining both within the same analysis section.
-
Results - Gender Distribution: The gender distribution between rUKA and cUKA groups could influence outcomes. Add a sentence acknowledging this as a potential confounding factor.
-
Results - Statistical Significance: When reporting mean differences, ensure confidence intervals (CIs) accompany the statistical results to reinforce precision, e.g., "Mean difference = 6.02 (95% CI: -0.07 to 12.1), p = 0.05."
-
Results - Range of Motion: Highlight the clinical relevance of the minimal ROM difference at various follow-up points to clarify its impact on patient functionality.
-
Results - Table Interpretation (Table 2): Adding an interpretive summary following each table may improve accessibility and facilitate understanding for readers.
-
Results - Forest Plot Labels (Figures 4-10): Ensure that forest plots are labeled clearly to allow for easier cross-referencing with the text, improving clarity for readers.
-
Discussion - PROMs Interpretation (Line 780): Differentiate the types of PROMs, such as FJS-12 for joint awareness versus KSS for satisfaction and pain, to clarify their varying relevance and impact on the findings.
-
Discussion - Limitations of Current Literature : The majority of studies included are retrospective, which limits causal conclusions regarding rUKA versus cUKA. Acknowledge this as a limitation.
-
Discussion - Long-term Data : Add a call for future studies to provide long-term data comparing rUKA and cUKA outcomes, addressing the current gap in extended follow-up information.
-
Discussion - Patient Satisfaction : Patient satisfaction factors, such as expectations from robotic surgery, should be discussed, as they may influence subjective PROMs.
-
Limitations Section - Heterogeneity : While heterogeneity is noted, the reasons are not fully discussed. Acknowledge that study design and patient population variations may contribute to heterogeneity in outcomes.
-
Limitations Section - Single Study Weight: When discussing negligible heterogeneity, clarify that a small number of studies or dominance of individual studies may obscure broader trends, affecting the reliability of findings.
-
Conclusion - Clinical Recommendations: Refine the clinical recommendations by specifying patient demographics or conditions that might particularly benefit from rUKA.
-
Supplemental Material - Search Strategy: Ensure that all search strategies and inclusion criteria are documented in the supplemental material to improve transparency and reproducibility.
Minor English editing required
Author Response
Dear Sir or Madam,
We have attached a Word document in which we addressed all of your suggestions.
Kind regards,
The author team

Reviewer 2 Report
Comments and Suggestions for Authors
Methodology Comments
Search Strategy Scope:
The authors utilized five databases and adhered to PRISMA guidelines. Were any relevant databases omitted (e.g., Embase)? Could expanding the search improve the comprehensiveness?
Were specific keywords or MeSH terms optimized for capturing all relevant studies?
Inclusion and Exclusion Criteria:
The inclusion criteria focus on comparative studies but exclude surgical technique reports. Could valuable data regarding robotic-assisted UKA technique refinements be overlooked?
How was the heterogeneity of study designs (retrospective vs. prospective) addressed in terms of potential bias?
Subgroup Analysis:
Were subgroup analyses pre-specified in the protocol or conducted post hoc? Could this impact the reliability of subgroup conclusions?
Was any sensitivity analysis conducted to assess the impact of lower-quality studies on the meta-analysis results?
I² Statistic Management:
The protocol mentions random-effects modeling. For studies with an I² > 75%, were additional approaches (e.g., meta-regression) used to explore sources of heterogeneity?
Results Comments
Patient-Reported Outcomes (PROMs):
The results highlight better FJS-12 outcomes for rUKA. Were PROMs standardized across studies, and how were differences in measurement tools handled?
How were missing data in PROMs addressed in individual studies included in the analysis?
Complication Rates:
The findings indicate significantly lower aseptic loosening rates with rUKA. Were all complications uniformly defined and consistently reported across studies?
Could differences in follow-up duration between studies bias these complication rate comparisons?
Revision Rates:
The OR for all-cause revisions shows a significant benefit for rUKA. Were reasons for revisions consistently categorized (e.g., surgeon error, implant issues)?
Did the meta-analysis consider differences in surgeon experience between groups as a potential confounder?
Heterogeneity and Confidence Intervals:
With an I² approaching the high threshold in some outcomes, how was this heterogeneity justified or mitigated in the results?
Were confidence intervals for all metrics sufficiently narrow to draw robust conclusions?
Author Response
Dear Sir or Madam,
We have attached a Word document in which we address all of your suggestions.
Thank you for dedicating your time to revise our paper.
Kind regards,
The author team

Reviewer 3 Report
Comments and Suggestions for Authors
Dear authors,
Congratulations on your research, the following comments are suggestions that could help improve your manuscript.
In the introduction section it might be appropriate to provide more information on knee arthroplasties and especially on the pathological state that leads to surgical replacement.
What characteristics must the patient have to undergo this surgical intervention? Do patient characteristics influence the choice of conventional surgical technique? Justify it.
Could you indicate what the main patient-reported outcome measures are?
The methodology section is appropriate and correctly includes the search strategy. The criteria indicated seem adequate. It is noteworthy that the PROSPERO registration code has been included, thanks.
In the tables in the results section, include a table footer explaining the acronyms used.
The discussion could be improved by including more content and references. What is the relationship between the surgical procedure and the subsequent rehabilitation process? Is it possible to find changes in the results if the post-surgical physiotherapy treatment is modified?
Author Response
Dear Sir or Madam,
Thank you for taking the time to revise our manuscript.
We have attached a Word document in which we address all of your suggestions.
Kind regards,
The author team
